# Analysis of the *Drosophila* Ajuba LIM protein defines functions for distinct LIM domains

**Cordelia Rauskolb, Ahri Han, Elmira Kirichenko, Consuelo Ibar, Kenneth D. Irvine** [ORCID]*

Waksman Institute and Department of Molecular Biology and Biochemistry, Rutgers University, Piscataway, NJ, United States of America

* irvine@waksman.rutgers.edu

**Data Availability Statement:** All relevant data are within the paper and its Supporting Information files.

**Funding:** This research was supported by National Institutes of Health (www.nih.gov) grant

## Abstract

The Ajuba LIM protein Jub mediates regulation of Hippo signaling by cytoskeletal tension through interaction with the kinase Warts and participates in feedback regulation of junctional tension through regulation of the cytohesin Steppke. To investigate how Jub interacts with and regulates its distinct partners, we investigated the ability of Jub proteins missing different combinations of its three LIM domains to rescue *jub* phenotypes and to interact with α-catenin, Warts and Steppke. Multiple regions of Jub contribute to its ability to bind α-catenin and to localize to adherens junctions in *Drosophila* wing imaginal discs. Co-immunoprecipitation experiments in cultured cells identified a specific requirement for LIM2 for binding to Warts. However, in vivo, both LIM1 and LIM2, but not LIM3, were required for regulation of wing growth, Yorkie activity, and Warts localization. Conversely, LIM2 and LIM3, but not LIM1, were required for regulation of cell shape and Steppke localization in vivo, and for maximal Steppke binding in co-immunoprecipitation experiments. These observations identify distinct functions for the different LIM domains of Jub.

## Introduction

A wide range of functions have been identified for the Ajuba family of LIM domain proteins, including effects on transcription, intercellular signaling pathways, microRNA processing, and the cytoskeleton [1, 2]. Ajuba family proteins have also been ascribed a correspondingly wide range of cellular locations, including cytoplasm, nucleus, centrosomes, adherens junctions, focal adhesions, and P bodies [3–11]. Structurally, they are characterized by the presence of three C-terminal LIM domains, which are modular protein interaction motifs. While mammals have three Ajuba family proteins: AJUBA, WTIP, and LIMD1, *Drosophila* has only one, the Ajuba LIM protein (Jub). Jub is best known for its role in regulating Hippo signaling [12, 13], but it also influences cellular and cytoskeletal organization [11, 14, 15]. How the multiple functions of Ajuba family proteins are coordinated is not well understood.

Diverse groups of LIM domain proteins can localize to the actin cytoskeleton, adherens junctions (AJ) or focal adhesions (FA) when there is tension in the actin cytoskeleton [16, 17]. Localization of Ajuba family proteins to AJ also requires α-catenin, a key component of AJ, which can bind to both ß-catenin and F-actin [4, 18]. Tension in the actin cytoskeleton can induce conformational changes in α-catenin that enable association with Ajuba proteins and

GM131748 to KDI. The funders had no role in study design, data collection and analysis, decision to publish, or preparation of the manuscript.

**Competing interests:** The authors have declared that no competing interests exist.

with Vinculin at AJ, although they each bind distinct regions of α-catenin [4, 18–25]. Binding between Vinculin and α-catenin is well characterized, but how Ajuba proteins recognize α-catenin is less clear. Binding of Jub was mapped to the N2 alpha helical bundle of α-catenin [20], but the region of Jub responsible for this binding to α-catenin was not defined. Examination of association between mammalian Ajuba and α-catenin identified the LIM domains as primarily responsible for binding to α-catenin [4].

One key function of Ajuba family proteins is physical interaction with and inhibition of Warts (Wts) kinases [13, 18, 19, 26–30]. Wts (LATS1 and LATS2 in mammals) is the central kinase of the Hippo signaling network [31]. The Hippo signaling network controls organ growth and cell fate in a wide range of animals, and when dysregulated, can contribute to cancer [31, 32]. Hippo signaling mediates its effects largely through regulation of the transcriptional co-activator protein Yorkie (Yki, YAP1 and TAZ in mammals) [33]. Yki is inhibited by Hippo signaling through phosphorylation by Wts [34, 35]. Hippo signaling integrates diverse upstream inputs, including cytoskeletal tension [31, 36]. One mechanism through which cytoskeletal tension modulates Hippo signaling involves the tension-dependent recruitment and inhibition of Wts at AJ, which is mediated by Jub. This was first discovered in *Drosophila* wing imaginal discs [18], but is conserved in mammalian cells, as LIMD1 can recruit and inhibit LATS kinases at AJ in MCF10A cells when there is tension in the actin cytoskeleton [19, 37, 38]. The recruitment of Wts to AJ by Jub sequesters it from upstream activators of Hippo signaling [39].

In addition to regulating Hippo signaling through regulation of Wts, Jub also has effects on cellular organization, cell shape and myosin distribution that are not linked to effects on the Hippo pathway [14, 15]. These phenotypes associated with loss or mutation of *jub* in embryos or imaginal discs are similar to those observed upon loss or mutation of *steppke* (*step*) [14, 40], a member of a family of proteins (cytohesins) that function as GEFs for Arf family G proteins [41]. Step modulates tension at adherens junctions, possibly by regulating Rho or WAVE [14, 40, 42, 43]. Jub is required for localization of Step to AJ [14], but the basis for this recruitment hasn't been determined.

To better understand how Jub interacts with multiple partners to carry out its distinct functions, we investigated the contributions of different regions of the Jub protein to its in vivo functions and its interactions with distinct proteins. Multiple regions of Jub contribute to its ability to associate with α-catenin, and to its localization to AJ in wing imaginal discs. The second LIM domain (LIM2) is specifically required for association with Wts in co-immunoprecipitation experiments, but both LIM1 and LIM2 are required for regulation of growth, Yki activity, and Wts localization in wing discs. Conversely, LIM2 and LIM3, but not LIM1, are required for regulation of cell shape and Step localization in wing discs, and for maximal Step binding in co-immunoprecipitation experiments. Our results thus identify distinct functions for different LIM domains of Jub and provide a foundation for further investigations of how cytoskeletal tension regulates Wts and Step through Jub.

## Results

### Ajuba LIM domains have conserved differences in sequence

Ajuba family proteins have a long N-terminal region (preLIM region) followed by three LIM domains. The preLIM region is not highly conserved, and computational analysis [44] of the Jub preLIM sequence indicates that much of it has a high probability of forming an intrinsically disordered region (Figs 1A and S1). Conversely, the LIM domains are highly conserved, and are not predicted to be disordered. Moreover, comparing Jub to the three mammalian Ajuba family protein members, each of the LIM domains is more similar to the corresponding

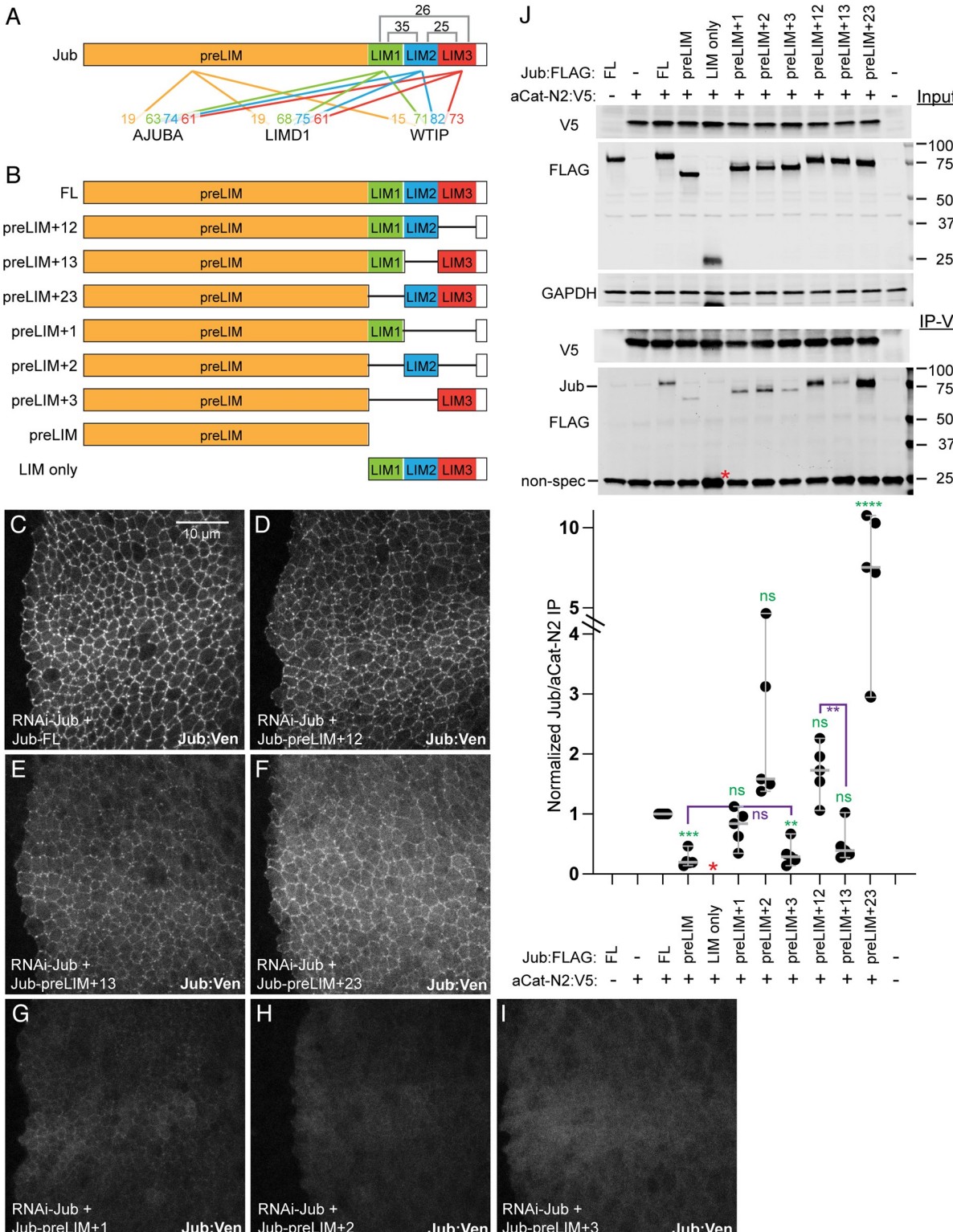

**Fig 1. Expression of Jub constructs and their association with α-catenin.** A) Linear schematic of Jub protein, with preLIM, LIM1, LIM2, and LIM3 regions colored. Numbers above indicate percent sequence identity between LIM domains of Jub. Numbers below indicate percent sequence identity for each of the four colored regions to corresponding regions of AJUBA, LIMD1, and WTIP. B) Schematics of the different Jub protein constructs used. C-I) Expression of UAS-msVenus-tagged Jub constructs in wing discs, expressed under *hh-Gal4* with endogenous Jub knocked down by RNAi. All images were captured under identical conditions, to illustrate relative expression, and are at the

same magnification (scale bar = 10 μm). C) *hh-Gal4 UAS-Dcr2 UAS-RNAi jub UAS-Jub-FL:msVenus*. D) *hh-Gal4 UAS-Dcr2 UAS-RNAi jub UAS-Jub-preLIM+12:msVenus*. E) *hh-Gal4 UAS-Dcr2 UAS-RNAi jub UAS-Jub- preLIM+13:msVenus*. F) *hh-Gal4 UAS-Dcr2 UAS-RNAi jub UAS-Jub- preLIM+23:msVenus*. G) *hh-Gal4 UAS-Dcr2 UAS-RNAi jub UAS-Jub- preLIM+1:msVenus*. H) *hh-Gal4 UAS-Dcr2 UAS-RNAi jub UAS-Jub- preLIM+2:msVenus*. I) *hh-Gal4 UAS-Dcr2 UAS-RNAi jub UAS-Jub- preLIM+3:msVenus*. J) Results of co-immunoprecipitation experiments between FLAG-tagged Jub constructs and the N2 domain of α-catenin, tagged with V5. Top three gels show western blots of cell lysates transfected to express the indicated proteins, using antibodies indicated at left. Bottom two gels show western blots on material precipitated with anti-V5 beads. In the lower blot, Jub:FLAG construct bands are at top and a non-specific band is at bottom, except for LIM only (red asterisk), which overlaps the non-specific band. Scatter plot at bottom shows quantitation of relative Jub/α-catenin-N2 band intensity in immunoprecipitates, normalized to the results for full length Jub, from five biological replicates, with mean and 95% confidence intervals indicated by gray bars. Statistical comparisons to the values for full length Jub are indicated in green, selected additional comparisons are indicated in purple.

LIM domain within other Ajuba family proteins than it is to other LIM domains within the same protein. For example, each of the LIM domains of Jub has sequence identities of 25–35% to the other LIM domains of Jub, but sequence identities of 61–82% to their corresponding LIM domains with human AJUBA, LIMD1, or WTIP (Fig 1A). This pattern of sequence similarity suggests that each of the LIM domains has distinct functions and that these distinct functions are evolutionarily conserved.

### Localization of Jub deletion constructs in wing discs

To assess requirements for Jub LIM domains, we took advantage of a series of Jub deletion constructs (Fig 1B) that were previously used to assess the requirements for different regions of Jub in its junctional localization in the early embryo [15]. We assessed requirements for different regions of Jub in wing imaginal discs, which have served as a model for studies of both Hippo signaling and tissue mechanics [45]. When a full length Jub:msVenus construct (Jub-FL) was expressed in wing discs under UAS control using a *hedgehog* (*hh*) driver (*hh-Gal4*), we found that it localizes predominantly to AJ, and often concentrates in bright puncta (Fig 1C). This resembles the pattern previously observed using a genomic Jub:GFP construct [18]. Constructs with two of the three LIM domains retained were also readily detected on AJ and concentrated in relatively bright puncta, although the level of junctional localization appears reduced compared to Jub-FL (Fig 1C–1F). Constructs with a single LIM domain retained were detected at even lower levels, and while Jub-preLIM+1 was weakly junctional, Jub-preLIM+2 and Jub-preLIM+3 were predominantly cytoplasmic (Fig 1G–1I). The more diffuse localization of the single LIM domain constructs likely contributes to the visibly reduced signal.

### Association of Jub constructs with the α-catenin N2 domain

Jub can be co-immunoprecipitated with α-catenin in lysates of wing imaginal discs [18]. In co-immunoprecipitation experiments conducted in S2 cells, which lack AJ, the N2 domain of α-catenin (αcat-N2) was found to associate with Jub in co-immunoprecipitation experiments [20]. To determine which region of Jub is responsible for this association, we assayed the ability of an αcat-N2 construct to co-immunoprecipitate different Jub deletion constructs. For these experiments we used constructs containing the same regions of Jub as for the in vivo localization experiments, but with the msVenus tag replaced by a C-terminal FLAG tag.

These co-immunoprecipitation experiments revealed some association of αcat-N2 with both the pre-LIM and LIM domain regions, although the association with the preLIM region was substantially weaker than for full length Jub (24% of full-length levels, Fig 1J), and association with the LIM domain only construct could not be accurately quantified due to overlap with a non-specific background band on the blots. The constructs that included both the pre-LIM region and LIM1 or LIM2 each associated with αcat-N2 similar to full length Jub (Fig 1J), whereas preLIM+LIM3 construct co-immunoprecipitation was weak, and similar to that of

preLIM alone. Amongst constructs with two LIM domains, preLIM+13 co-immunoprecipitation (50% of full length) was intermediate between that of preLIM+LIM1 and preLIM+LIM3 binding, while Jub-preLIM+12 co-immunoprecipitation (170% of full length) was slightly greater, but the differences as compared to full length were not significant. Jub-preLIM+23 co-immunoprecipitation with αcat-N2 was substantially greater (780% of full length) (Fig 1J). Despite this more robust co-immunoprecipitation, Jub-preLIM+23 did not show stronger association with AJ in vivo (Fig 1C and 1F), but we note that the interactions detected by co-immunoprecipitation reflect association with a fragment of α-catenin rather than the full length protein, and occur in the cytoplasm rather than at AJ.

## Requirements for Jub LIM domains for development of adult wings

To examine the biological activity of Jub deletion constructs during wing development, we assayed their ability to rescue the adult wing phenotype associated with RNAi-mediated *jub* knockdown. This was accomplished by using a UAS-controlled hairpin construct to knock down endogenous *jub* expression, and then rescuing this knockdown using UAS-Jub constructs that don't contain the sequence targeted by the shRNA. Knockdown of *jub* under *nub-Gal4* control reduces wing size (wings were 54% of *nub-Gal4* control size, Fig 2A, 2B and 2L), consistent with earlier studies [12, 13]. This reduction in wing size is largely rescued by co-expression of full length UAS-Jub (wings were 91% of *nub-Gal4* control size, Fig 2C and 2L). Examination of rescue with Jub deletion constructs revealed that each of the preLIM plus single LIM domain constructs provided little or no rescue of *jub* RNAi wing size (wings were 57, 61, and 60% of control size)(Fig 2G–2I and 2L). However, when examining the Jub constructs containing two LIM domains, differences emerged. Deletion of only LIM2 prevented rescue of *jub* RNAi (wings were 56% of control size, Fig 2E and 2L), but constructs with only LIM1 deleted partially rescued *jub* RNAi (wings were 76% of control size, Fig 2F and 2L). Deletion of LIM3 actually increased wing size as compared to rescue with full length Jub (wings were 97% of *nub-Gal4* control size, Fig 2D and 2L). These observations suggest that LIM2 is essential for Jub's ability to inhibit Wts activity in the wing, LIM1 also contributes to this activity, and LIM3 is not required and might to some degree even antagonize it.

To further evaluate the possibility that deletion of LIM3 increases Jub's activity in inhibiting Wts, we compared the consequences of expressing Jub-FL and Jub-preLIM+12 in the absence of endogenous *jub* knock down (Fig 2J–2L). Full length Jub over-expression did not significantly alter wing size (wings were 101% of *nub-Gal4* control). Over-expression of Jub-preLIM +12 slightly increased wing size (wings were 105% of *nub-Gal4* control), but the difference in wing size between these two genotypes was not statistically significant.

## Requirements for Jub LIM domains in regulation of Yki activity

The decreases in wing size observed with many of the LIM domain deletion constructs suggests that they are defective in regulation of Hippo signaling. To investigate this possibility, we examined the expression of Yki target genes in wing discs. UAS-RNAi-*jub* and UAS-Jub constructs were expressed together in posterior cells under *hh-Gal4* control, so that anterior cells could serve as an internal control. Consistent with earlier studies [12, 13], RNAi-mediated knockdown of *jub* leads to reduction in expression of *ex-lacZ* (Figs 3A, 3F and S1), a well-characterized transcriptional reporter for *ex*, which is a direct target of Yki [46]. This reduction is rescued by expression of full length UAS-Jub (Fig 3B and 3F). Each of the Jub constructs that included only a single LIM domain with the preLIM region failed to rescue *ex-lacZ* expression (Figs 3F and S2), and these were not examined further. Amongst the Jub constructs with two LIM domains, Jub-preLIM+13 completely failed to rescue the reduction of *ex-lacZ* expression,

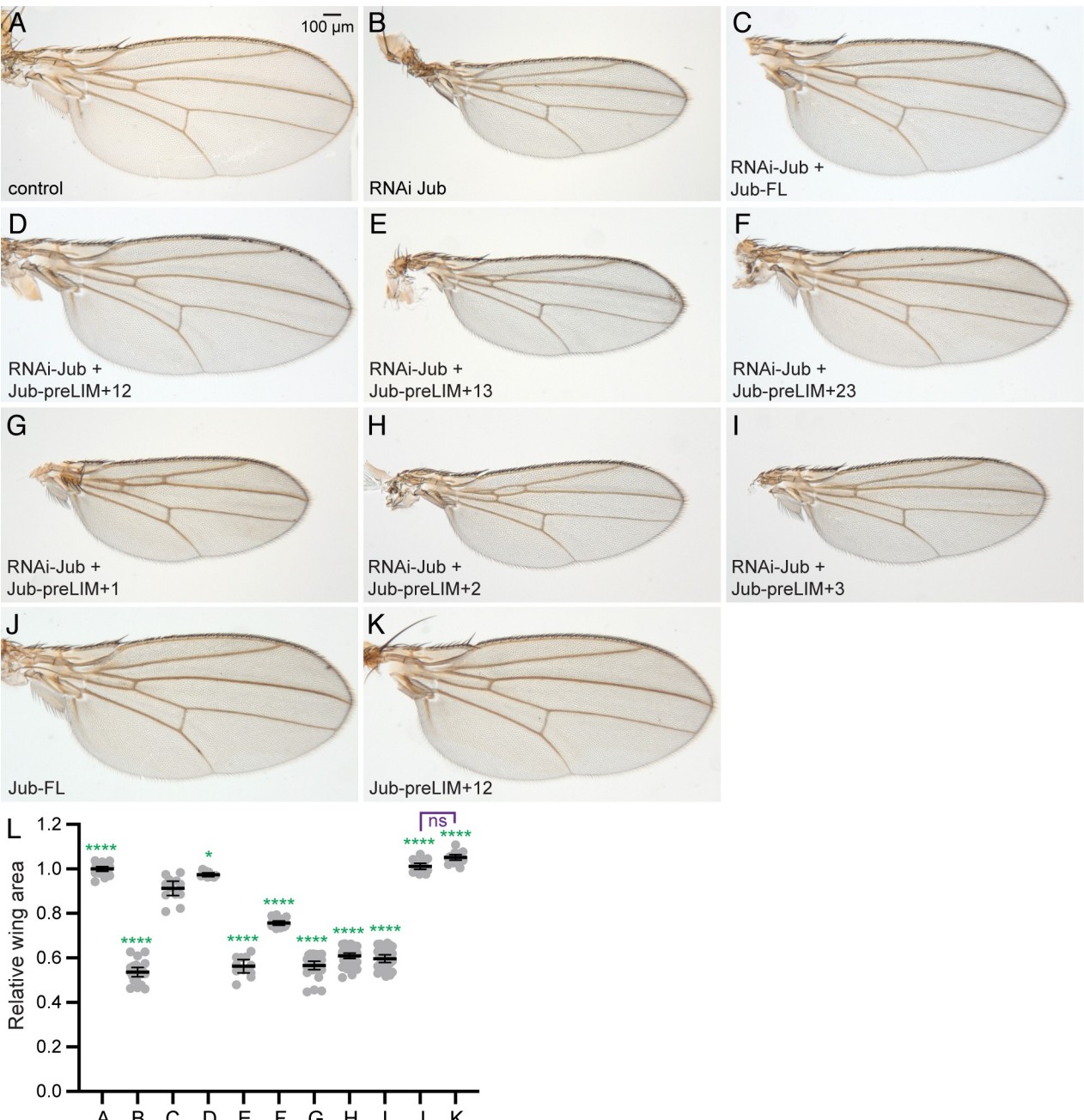

**Fig 2. Rescue of adult wings by Jub deletion constructs.** A-K) Representative adult wings from male flies expressing *nub-Gal4 UAS-Dcr2* and A) control. B) *UAS-RNAi-jub*. C) *UAS-RNAi-jub UAS-Jub-FL:msVenus*. D) *UAS-RNAi-jub UAS-Jub-preLIM+12:msVenus*. E) *UAS-RNAi jub UAS-Jub-preLIM+13:msVenus*. F) *UAS-RNAi jub UAS-Jub- preLIM+23:msVenus*. G) *UAS-RNAi jub UAS-Jub- preLIM+1:msVenus*. H) *UAS-RNAi jub UAS-Jub- preLIM+2:msVenus*. I) *UAS-RNAi jub UAS-Jub- preLIM+3:msVenus*. J) *UAS-Jub-FL:msVenus*. K) *UAS-Jub-preLIM+12:msVenus*. All wings are shown at the same magnification (scale bar = 100 μm). L) Scatter plot showing quantitation of wing area, normalized to the mean wing area in control flies, with mean and 95% ci indicated by black bars. Letters identify the genotypes as in the panels above, number of wings measured is A) 21, B) 22, C) 13, D) 10, E) 11, F) 18, G) 29, H) 43, I) 31, J) 17, K) 17. The significance of differences in expression levels as compared to *UAS-RNAi-jub UAS-Jub-FL:msVenus* are indicated by green asterisks, comparison between J and K is indicated in purple.

whereas Jub-preLIM+12 fully rescued *ex-lacZ* expression (Fig 3C–3F). *ex-lacZ* expression in Jub-preLIM+23 was slightly above that detected in Jub-preLIM+13 wing disc cells. To further assay the ability of these constructs with two LIM domains to rescue Yki activity in *jub* RNAi

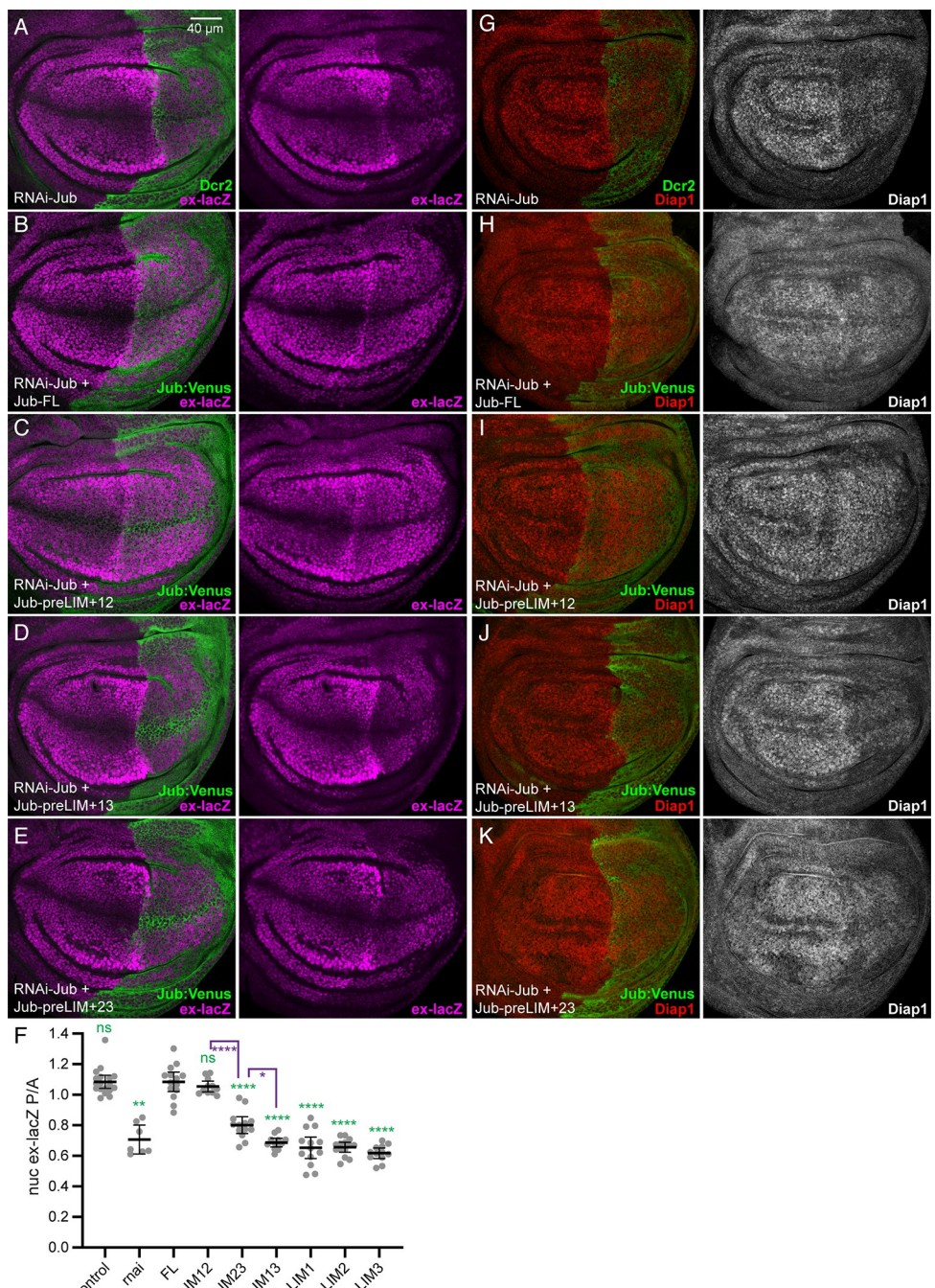

**Fig 3. Rescue of Yki target gene expression by Jub deletion constructs.** A-E) Representative wing discs from larvae expressing *ex-lacZ* (magenta), *hh-Gal4 UAS-Dcr2* and A) *UAS-RNAi-jub*. B) *UAS-RNAi-jub UAS-Jub-FL:msVenus*. C) *UAS-RNAi-jub UAS-Jub-preLIM+12:msVenus*. D) *UAS-RNAi jub UAS-Jub-preLIM+13:msVenus*. E) *UAS-RNAi jub UAS-Jub-preLIM+23:msVenus*. All discs are shown at the same magnification (scale bar = 40 μm). F) Scatter plot showing quantitation of *ex-lacZ* levels in the posterior wing pouch as compared to the levels in the anterior wing pouch, with mean and 95% ci indicated by black bars. Genotypes are indicated by abbreviations at bottom. Representative examples of *ex-lacZ* expression in control wing discs and wing discs from animals expressing single LIM domain Jub constructs are in S2 Fig. Numbers of wing discs measured is control) 18, jub RNAi) 7, *Jub-FL*) 14, *Jub-preLIM+12*) 11, *Jub-preLIM+23*) 13, *Jub-preLIM+13*) 12, *Jub-preLIM+1*) 13, *Jub-preLIM+2*) 14, *Jub-preLIM+3*) 12. The significance of differences in expression levels as compared to *UAS-RNAi-jub UAS-Jub-FL:msVenus* are indicated by green asterisks, additional selected comparisons are in purple. G-K) Representative wing discs stained for Diap1 (red/white) from larvae expressing *hh-Gal4 UAS-Dcr2* and A) *UAS-RNAi-jub*. B) *UAS-RNAi-jub UAS-Jub-FL: msVenus*. C) *UAS-RNAi-jub UAS-Jub-preLIM+12:msVenus*. D) *UAS-RNAi jub UAS-Jub-preLIM+13:msVenus*. E) *UAS-RNAi jub UAS-Jub-preLIM+23:msVenus*. All discs are shown at the same magnification as for *ex-lacZ* stains.

cells, we also examined expression of the Yki target gene *Diap1*, using an anti-Diap1 antibody. As for *ex-lacZ* expression, Jub-preLIM+12 rescued Diap1 expression, whereas Jub-preLIM+23 and Jub-preLIM+13 exhibited partial or no rescue of Diap1 expression (Fig 3G–3K), respectively.

## Requirements for Jub LIM domains in regulation of Wts localization

Jub affects Yki activity by regulating Wts activity and localization [13, 18, 39]. To determine whether Jub deletion constructs influence Wts localization, and whether their influence on Wts correlates with effects on Yki activity, we used a genomic Wts:V5 construct to assess Wts localization in wing disc cells expressing Jub deletion constructs. RNAi of *jub* leads to loss of Wts from AJ (Fig 4A and 4F), consistent with earlier studies of *jub* mutant clones [18]. This loss of Wts from AJ is rescued by expression of full length Jub (Fig 4B and 4F). Examination of each of the Jub constructs with two LIM domains revealed that Jub-preLIM+12 fully rescued junctional localization of Wts, whereas Jub-preLIM+13 completely failed to rescue junctional Wts and Jub-preLIM+23 provided a weak, partial rescue of Wts localization (Fig 4C–4F). The correlation between the effects of Jub constructs on Wts localization and their effects on Yki activity support the inference that recruitment of Wts to AJ is central to its regulation by Jub.

## Association of Jub constructs with Wts

To map regions of Jub that are responsible for its physical association with Warts, we co-expressed FLAG-tagged Jub constructs together with V5-tagged Wts in cultured *Drosophila* S2 cells and then assayed for their association by co-immunoprecipitation. Co-immunoprecipitation of full-length Jub with Wts was readily detected by western blotting (Fig 5A), consistent with prior studies of interactions between Jub and Wts, and between mammalian Ajuba proteins and Lats kinases [13, 26, 29, 30]. When Jub deletion constructs were co-expressed with Wts, all constructs that included LIM2 (LIM only, preLIM+2, preLIM+12, preLIM+23) co-immunoprecipitated with Wts at levels similar to or even greater than full length Jub, whereas all constructs lacking LIM2 exhibited little or no co-precipitation (at least 7-fold lower that full length Jub) (Fig 5A). Together, these observations indicate that LIM2 of Jub makes the primary contribution to association with Wts, whereas the preLIM region, LIM1, and LIM3 make little or no contribution to Wts binding, at least under the conditions of this assay.

## Influence of Jub constructs on cell shape

Jub is also required for junctional localization of the cytohesin Step [14]. Loss of Jub or Step can influence the distribution of myosin, epithelial cell shapes, and epithelial cell organization in both wing discs and embryos [14, 15, 40]. To investigate the potential impact of different Jub isoforms on Step function in wing discs, we first examined cell shapes and myosin localization in wing discs expressing different Jub constructs, together with *jub* RNAi. While myosin localization is aberrant in cells lacking Jub, it was difficult to clearly assess the rescue of this phenotype with the various Jub transgenes (Fig 6A–6E). Instead, the most obvious and quantifiable difference between cells in *jub* RNAi wing discs, and cells in *jub* RNAi wing discs rescued by expression of full length Jub, was the distribution of cell shapes, which tend to be more isometric in rescued discs, and more anisometric in *jub* RNAi discs. To quantify this, wing disc cells were segmented based on E-cadherin (Ecad) staining, and cell eccentricity was measured in the posterior compartment of the wing pouch, where these constructs were expressed under *hh-Gal4* control. Eccentricity is a geometric description of shape, formally eccentricity = c/a Where, c = distance from the center to the focus; a = distance from the center to the vertex. A circle has an eccentricity of 0, a line has an eccentricity of 1, and an ellipse has

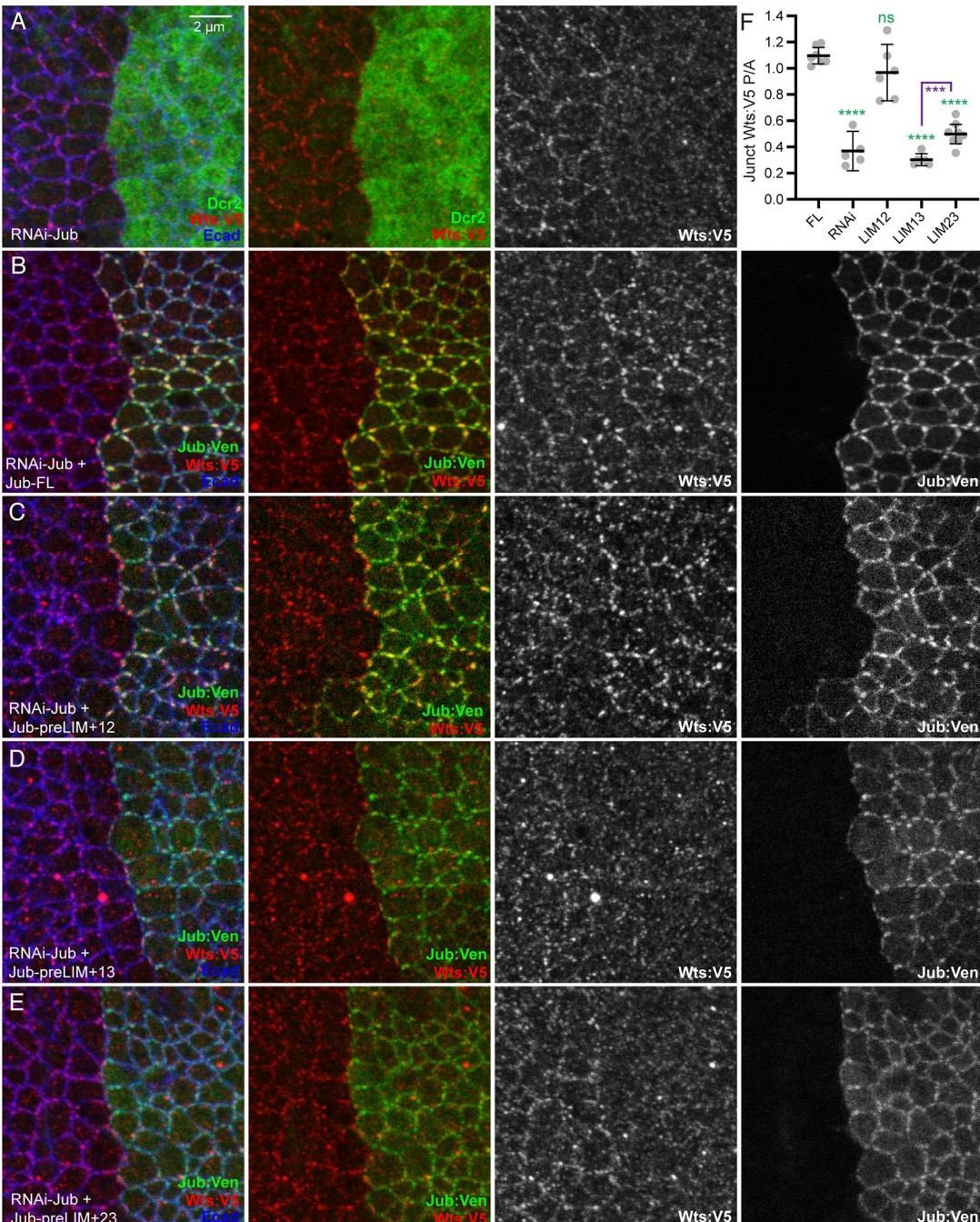

**Fig 4. Rescue of junctional Wts by Jub deletion constructs.** A-E) Representative wing discs from larvae expressing genomic tagged Wts:V5 (red), *hh-Gal4 UAS-Dcr2* and A) *UAS-RNAi-jub*. B) *UAS-RNAi-jub UAS-Jub-FL:msVenus*. C) *UAS-RNAi-jub UAS-Jub-preLIM+12:msVenus*. D) *UAS-RNAi jub UAS-Jub-preLIM+13:msVenus*. E) *UAS-RNAi jub UAS-Jub-preLIM+23: msVenus*. All discs are shown at the same magnification (scale bar = 2 μm). F) Scatter plot showing quantitation of Wts:V5 levels on AJ (defined by Ecad) in posterior cells as compared to the levels in anterior cells, with mean and 95% ci indicated by black bars. Genotypes are indicated by abbreviations at bottom. Numbers of wing discs measured are *Jub-FL*) 7, *jub* RNAi) 5, *Jub-preLIM+12*) 6, *Jub-preLIM+13*) 6, *Jub-preLIM+23*) 8. The significance of differences in expression levels (ANOVA) as compared to *UAS-RNAi-jub UAS-Jub-FL:msVenus* are indicated by green asterisks, additional selected comparisons are in purple.

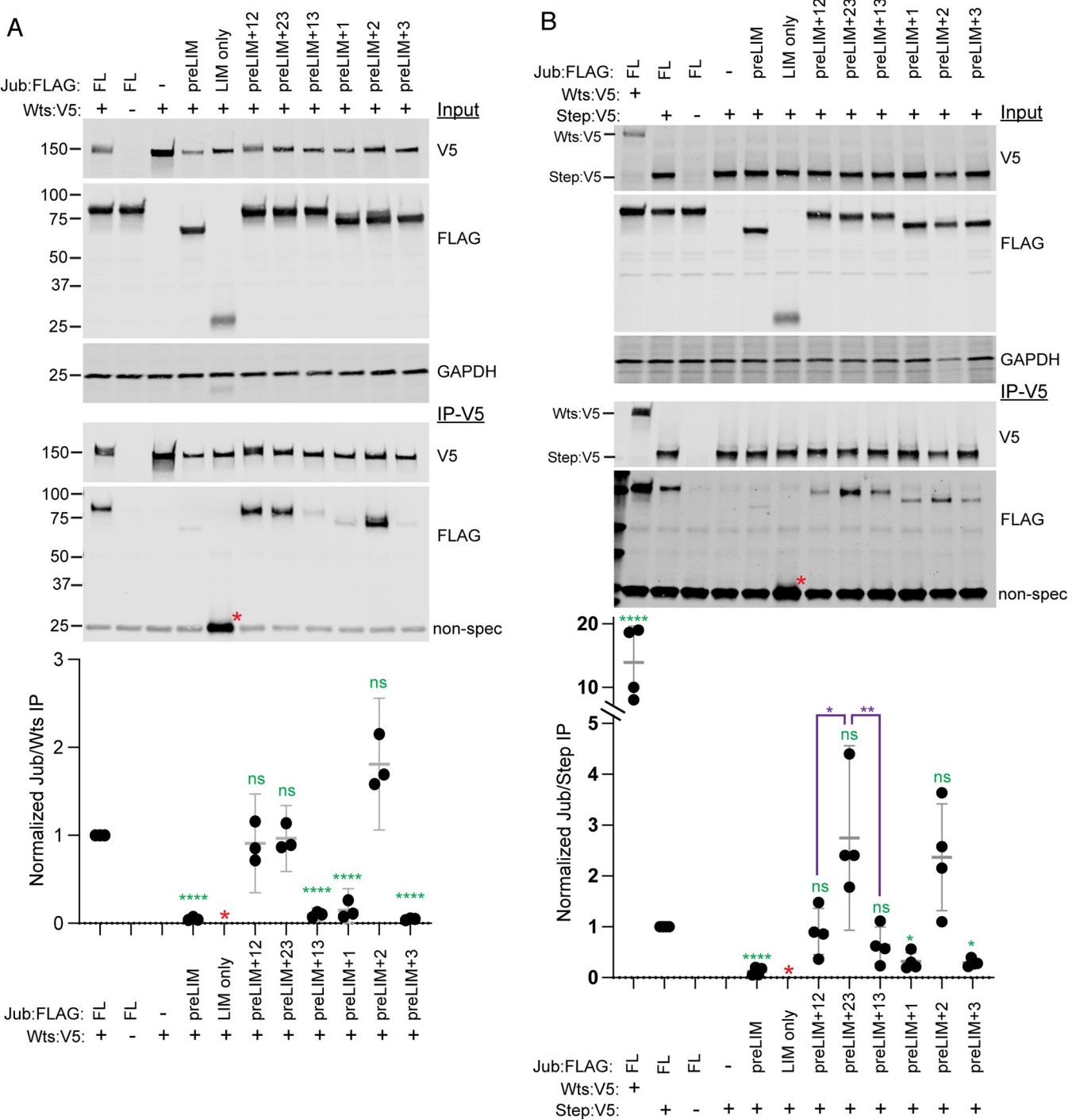

**Fig 5. Association of Jub with Wts and Step.** A) Results of co-immunoprecipitation experiments between FLAG-tagged Jub constructs and V5-tagged Wts. Top three gels show western blots of cell lysates transfected to express the indicated proteins, using antibodies indicated at right. Bottom two gels show western blots on material precipitated with anti-V5 beads. In the lower blot, Jub:FLAG construct bands are at top and a non-specific band is at bottom, except for LIM only (red asterisk), which overlaps the non-specific band. Scatter plot at bottom shows quantitation of relative Jub/Wts band intensity in immunoprecipitates, normalized to the results for full length Jub, from three biological replicates, with mean and 95% ci indicated by gray bars. Statistical comparisons (ANOVA) to the values for full length Jub are indicated in green. Quantitation of the LIM only band is not included because the signal cannot be separated from the background band. B) Results of co-immunoprecipitation experiments between FLAG-tagged Jub constructs and V5-tagged Step, with Wts:V5 also included for comparison of full length Jub binding. Top three gels show western blots of cell lysates transfected to express the indicated proteins, using antibodies indicated at right. Bottom two gels show western blots on material precipitated with anti-V5 beads. In the lower blot, Jub:FLAG construct bands are at top and a non-specific band is at bottom, except for LIM only (red asterisk), which overlaps the non-specific band. Scatter plot at bottom shows quantitation of relative Jub/

Step or Wts band intensity in immunoprecipitates, normalized to the results for full length Jub and Step, from four biological replicates, with mean and 95% ci indicated by gray bars. Statistical comparisons (ANOVA) to the values for full length Jub and Step are indicated in green, additional selected comparisons are in purple. Quantitation of the LIM only band is not included because the signal cannot be separated from the background band.

an eccentricity between 0 and 1, with more elongated ellipses having higher values (ie, closer to 1). While there is a broad distribution of cell eccentricity, it was significantly skewed towards higher values in *jub* RNAi wing discs as compared to discs rescued by expression of Jub-FL (Fig 6A, 6B and 6F). Expression of each of the Jub constructs with two LIM domains revealed that expression of Jub-preLIM+23 rescued this *jub* RNAi phenotype, restoring the distribution of eccentricity to be similar to that in discs rescued by full length Jub, whereas Jub-preLIM+12 or Jub-preLIM+13 did not fully rescue the cell eccentricity phenotype (Fig 6C–6F).

### Requirements for Jub LIM domains in regulation of Step localization

To determine whether the influence of Jub deletion constructs on cell shape correlates with influences on Step localization, we used a UAS-Step:mCherry construct [43] to examine Step localization in wing disc cells expressing Jub deletion constructs. When Step is expressed in wing discs, it localizes along AJ, and is often concentrated in puncta that overlap Jub puncta [14]. RNAi of *jub* throughout the wing under *nub-Gal4* control leads to loss of Step from adherens junctions (Fig 7E), consistent with earlier studies [14]. This loss of Step from AJ is rescued by co-expression of full length Jub (Fig 7A). Examination of each of the Jub constructs with two LIM domains revealed that Jub-preLIM+23 visibly rescued junctional localization of Step, whereas Jub-preLIM+12 and Jub-preLIM+13 did not (Fig 7B–7D). These observations imply that LIM2 and LIM3, but not LIM1, are required for Step recruitment to AJ in wing discs.

### Association of Jub constructs with Step

The physical basis for the requirement for Jub in recruitment of Step to AJ has not been described. We examined whether Jub and Step could be associated in the same complex when co-expressed in S2 cells, and found that Step could co-immunoprecipitate Jub, although it was substantially less effective at this than Wts (Fig 5B). To determine which regions of Jub are responsible for this association with Step, we assayed the ability of Step to co-immunoprecipitate different Jub deletion constructs. No significant association was detected with the preLIM region (co-precipitation signal was more than 8 times lower than with full length Jub), and the Jub-preLIM+1 or Jub-preLIM+3 constructs were co-precipitated at approximately 30% of the level of full length Jub. Conversely, the constructs containing two LIM domains, and the Jub-preLIM+2 construct, were all co-immunoprecipitated to at least 64% of the level of full length Jub, with the preLIM+LIM23 construct exhibiting the strongest co-immunoprecipitation.

### Discussion

Our analysis of the functional contributions of the three LIM domains of Jub has established that they have distinct roles. This is consistent with the evolutionary conservation of the distinct sequences of each LIM domain amongst Ajuba family proteins. Sequence analysis also reveals that a large part of the Jub preLIM region is likely to be intrinsically disordered, as has also been reported for LIMD1 [47]. Intrinsically disordered regions have been linked to formation of biological condensates [48]. Jub protein accumulates in puncta along AJ at sites of high tension [14, 18], and it is possible that phase separation contributes to the formation of visible Jub puncta.

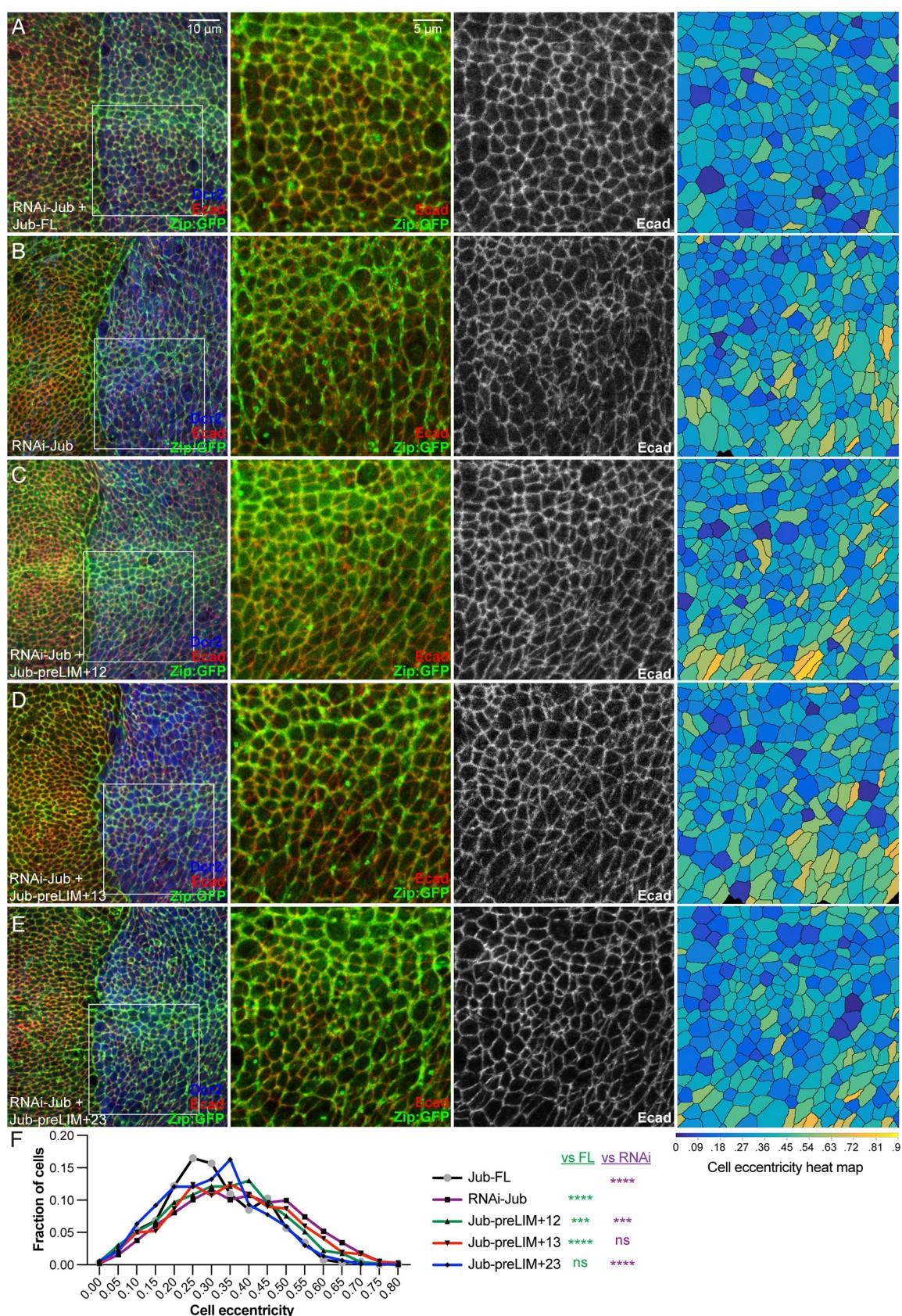

**Fig 6. Rescue of cell shape by Jub deletion constructs.** A-E) Representative wing discs from larvae expressing GFP-tagged myosin II heavy chain (Zipper, Zip:GFP, green), *hh-Gal4 UAS-Dcr2* and A) *UAS-RNAi-jub UAS-Jub-FL:msVenus.* B) *UAS-RNAi-jub.* C) *UAS-RNAi-jub UAS-Jub-preLIM+12:msVenus.* D) *UAS-RNAi jub UAS-Jub-preLIM+13:msVenus.* E) *UAS-RNAi jub UAS-Jub-preLIM +23:msVenus* and stained for Ecad expression (red/white). All discs are shown at the same magnification (scale bar = 10 μm). At far right segmented cells are colored according to eccentricity, which ranges from 0 (circular) to 1 (linear); scale for the heat map is at bottom. F) Plot showing the distribution of cell eccentricity in the posterior wing pouch of three wing discs each from animals of these five genotypes, as indicated. The significance of differences in the distribution (Kruskal-Wallis test with Dunn's test for multiple comparisons) as compared to that in *UAS-RNAi-jub UAS-Jub-FL:msVenus* are indicated by green asterisks, and the significance of differences in the distribution as compared to that in *UAS-RNAi-jub* are indicated by purple asterisks. Numbers of cells analyzed is *Jub-FL*) 879, jub RNAi) 905, *Jub-preLIM+12*) 1190, *Jub-preLIM+13*) 1035, *Jub-preLIM+23*) 1044.

Jub localizes to AJ, where it associates with α-catenin [11, 13, 18, 20]. Initial studies in mammalian cells found that association of AJUBA with α-catenin was primarily attributable to the LIM domains [4]. Our results with Jub also identify a key role for the LIM domains in α-catenin binding, and further suggest that while each of the LIM domains can contribute to α-catenin binding in co-immunoprecipitation experiments, LIM2 makes the greatest contribution and LIM3 the least. Examination of junctional localization of Jub constructs in wing discs is consistent with the idea that multiple LIM domains contribute to α-catenin association, as constructs with any two LIM domains exhibit significant junctional localization. However, the observation that multiple LIM domains are required for significant junctional localization in wing discs differs from studies in *Drosophila* embryos, where constructs with a single LIM domain, and even a preLIM-only construct, exhibited some junctional localization [15]. However, our results agree with studies in the embryo in finding that LIM1 and LIM2 make more significant contributions to junctional localization than LIM3. A structural understanding of α-catenin-Vinculin binding has been provided by co-crystals of fragments of these proteins [49]. We currently lack a structural understanding of how Ajuba family proteins associate with α-catenin, but studies of Jub deletion constructs suggest that multiple domains of Ajuba proteins contribute to this interaction.

A key function of *Drosophila* Jub is linking tension in the actin cytoskeleton to regulation of Hippo signaling by inhibition of Wts [18]. In examining the rescuing activity of different Jub constructs, we observed a correlation between effects on wing size, expression of Yki target genes, and localization of Wts to AJ, which supports the inference that recruitment of Wts to AJ is required for its inhibition by Jub. For example, Jub preLIM+LIM12 rescued each of these Jub functions as well as full length Jub, Jub preLIM+LIM13 completely failed to rescue all of these activities, and Jub preLIM+LIM23 provided partial rescue.

Co-immunoprecipitation experiments in S2 cells implicate LIM2 as being necessary and sufficient for association with Wts. However, in vivo, LIM2 is not sufficient for Wts regulation. If Jub needs to recruit Wts to AJ to inhibit it, then the failure of Jub preLIM+LIM2 to rescue *jub* phenotypes could be explained by the observation that this construct fails to localize to AJ in wing discs. However, Jub preLIM+LIM23 localizes to AJ, but provides only minimal regulation of Wts. Thus we infer that while LIM1 does not visibly contribute to Wts binding in S2 cells, it provides some additional function that contributes to junctional Wts recruitment and inhibition in wing discs.

The contributions of different LIM domains to regulation of Hippo signaling have also been examined for the mammalian LIMD1 protein, using co-immunoprecipitation experiments in cultured mammalian cells and expression of LIMD1 constructs in *Drosophila* wings [27]. In examining LIMD1-LATS2 association, Jagannathan et al [27] found that constructs lacking LIM2 could associate with LATS2, as long as LIM1 was present, which differs from our observations with Jub, which revealed an absolute requirement for LIM2. In rescue experiments, they reported that a construct missing only LIM3 could rescue wing size, but constructs

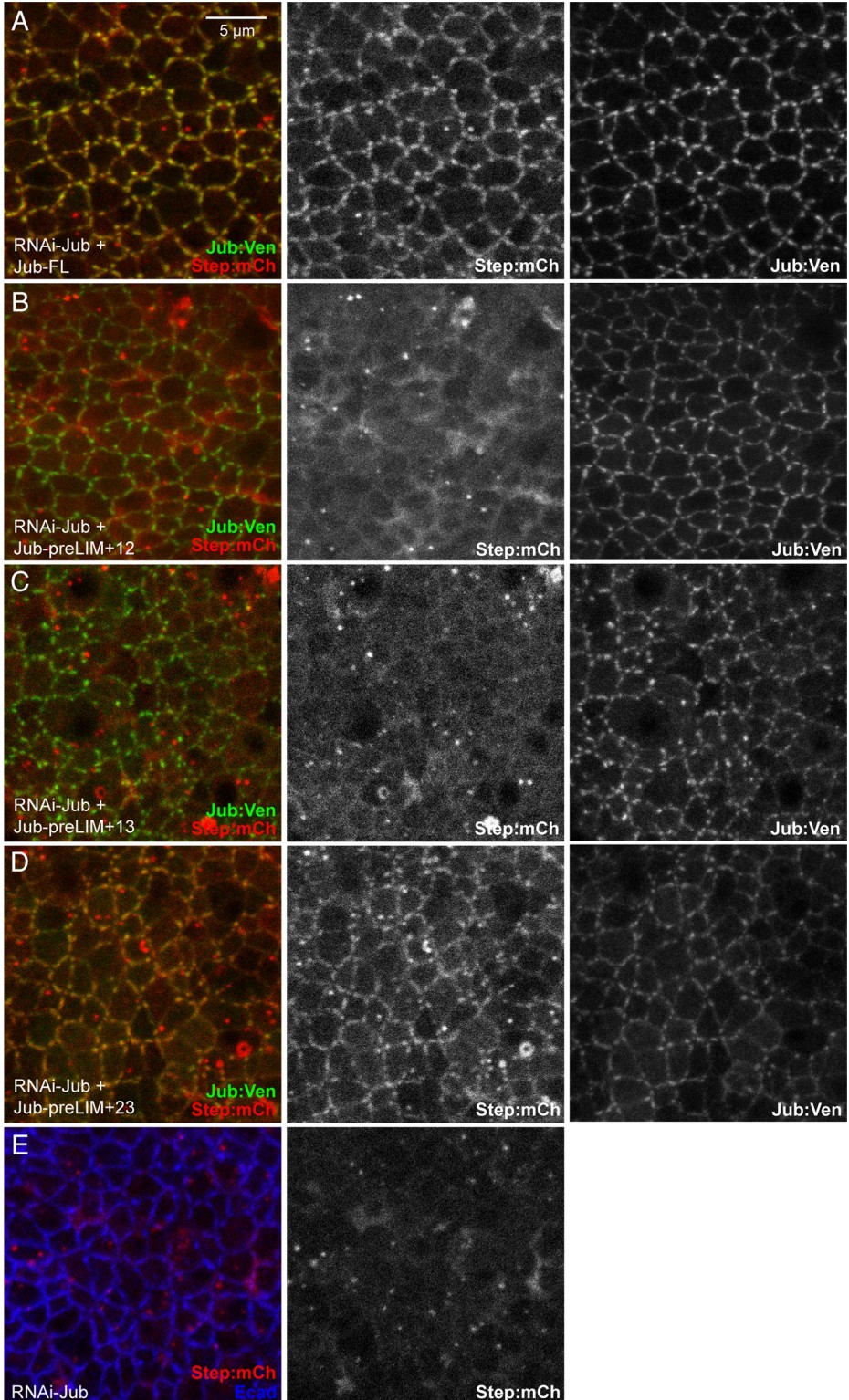

**Fig 7. Rescue of junctional Step by Jub deletion constructs.** A-E) Representative wing discs from larvae expressing Step:mCherry (red/white), *nub-Gal4 UAS-Dcr2* and A) *UAS-RNAi-jub UAS-Jub-FL:msVenus*. B) *UAS-RNAi-jub UAS-Jub-preLIM+12:msVenus*. C) *UAS-RNAi jub UAS-Jub-preLIM+13:msVenus*. D) *UAS-RNAi jub UAS-Jub-preLIM +23:msVenus*. E) *UAS-RNAi-jub*. All discs are shown at the same magnification (scale bar = 5 μm).

missing only LIM1 or only LIM2 could not, which is similar to our observations with Jub, except that they did not detect partial rescue of *jub* phenotypes with a construct similar to our Jub-preLIM+23. These differences might reflect real differences between Jub and LIMD1, or alternatively might stem from differences in experimental conditions.

Jub also modulates junctional tension and cell shape [14, 15]. We previously reported that Jub is required for junctional localization of Step in wing discs, but the nature of this requirement was unclear [14]. Co-immunoprecipitation experiments in S2 cells now suggest that Jub contributes to physical recruitment of Step to AJ, possibly by direct binding. The association of Jub with Step in S2 cells is though substantially weaker than the association of Jub with Wts. In vivo, an additional protein, Stepping stone (Sstn), is also required for recruitment of Step to cell junctions [14, 50], and one possible explanation for the relatively weak association of Jub with Step in our co-IP experiments could be the lack of additional Sstn.

In S2 cells, all three LIM domains appear to be able to contribute to association of Jub with Step. Nonetheless, amongst the constructs with two LIM domains, the strongest association was observed with the Jub-preLIM+23 construct, which is also the only Jub deletion construct that rescued Step recruitment in wing discs. The observation that Jub-preLIM+23 simultaneously rescues Step localization to AJ and the distribution of cell shapes also provides further support for the inference that Jub regulates cell shape by recruiting Step to AJ. Finally, we emphasize that the observations that Jub-preLIM+23 rescues Step regulation but not Wts regulation, while Jub-preLIM+12 rescues Wts regulation but not Step regulation, emphasizes that the defects in these proteins cannot be ascribed simply to reduced junctional levels of Jub, but rather must reflect specific and distinct contributions of LIM1 and LIM3 to Jub function. Thus, Jub coordinates biomechanical regulation of growth and of junctional mechanics through distinct LIM domains.

## Materials and methods

### *Drosophila* genetics

Fly crosses were set-up at 25˚C. After a day of egg-laying, the flies were transferred to a new vial and returned to 25˚C, whereas the vial with eggs and first instar larvae was transferred to 29˚C to attain maximal expression of *UAS-RNAi-jub* and the various Jub transgenes. Larvae from the 29˚C vial, at 96–120 hours after egg-laying, were dissected and further processed for gene expression and protein localization studies. For adult wing measurements, the vials were left at 29˚C until adult flies eclosed.

These stocks were used to set-up crosses with subsequent genotypes: *Oregon-R (control)*, *UAS-RNAi-jub [HMS02335]*, *UAS-RNAi-jub [HMS02335]; UAS-Jub-FL/TM6b, UAS-RNAi-jub [HMS02335]; UAS-Jub-preLIM+12:msVenus/TM6b, UAS-RNAi-jub [HMS02335]; UAS-Jub-preLIM+13:msVenus/TM6b, UAS-RNAi-jub [HMS02335]; UAS-Jub-preLIM+23:msVenus/ TM6b, UAS-RNAi-jub [HMS02335]; UAS-Jub-preLIM+1:msVenus/TM6b, UAS-RNAi-jub [HMS02335]; UAS-Jub-preLIM+2:msVenus/TM6b, UAS-RNAi-jub [HMS02335]; UAS-Jub-pre-LIM+3:msVenus/TM6*, and for Fig 2 only *UAS-Jub-FL:msVenus/TM6b* and *UAS-Jub-preLIM +12:msVenus/TM6b. UAS-RNAi-jub [HMS02335]* targets an untranslated region of the *jub* transcript, which is not included in the UAS-Jub transgenes.

The aforementioned stocks were used in the following assays: for assaying the adult wing size, we used *nub-Gal4 UAS-dcr2*; for effects on Yki-target gene expression, we used *ex-LacZ/ CyO-GFP; hh-Gal4 UAS-dcr2/ TM6b* and stained using antibodies detecting β-galactosidase or Diap1; to assay the effects of the different Jub deletion constructs on Wts and Step localization we used *yw P[acman]-Myc:Wts:V5[attP-2A]; hh-Gal4 UAS-dcr2/ TM6b* or *nub-Gal4 UAS-dcr2 UAS-Step:mCherry/ CyO*, respectively; to determine the effects on myosin localization and

cellular eccentricities we used *zip*:*GFP/ CyO; hh-Gal4 UAS-dcr2/ TM6b*, along with E-cadherin antibody staining. UAS-Jub constructs were obtained from J. Zallen [15], *UAS-Step*:*mCherry* is described in Lee and Harris (43), *P[acman]-Myc*:*Wts*:*V5[attP-2A]* is described in Cho et al (2006) [51].

## Wing measurements

Adult male wings were mounted in Gary's Magic Mountant [52]. Same magnification photographs were taken of at least 10 wings (average of 22) per genotype, and then digital images were traced using FIJI software to determine relative wing blade areas.

## Histology and imaging

Fixation and antibody staining of wing discs was performed as described in Rauskolb and Irvine [53], using 4% paraformaldehyde for 15 minutes. The following primary antibodies were used: Mouse anti-ß-gal (DSHB JIE7-c 1:400), Rat anti-E-cadherin (DSHB DCAD2-c 1:400), Rabbit anti-Dcr2 (Abcam ab4732, 1:1600), Mouse anti-Diap1 (gift of Bruce Hay, 1:200), Mouse anti-V5 (Invitrogen R960-25; preabsorb 1:10, then use 1:40). Secondary antibodies were purchased from Jackson ImmunoResearch Laboratories and Invitrogen. DNA was stained with Hoechst 33342 (Life Technologies).

Images were collected on a Leica SP8 confocal. For those experiments for which levels of expression amongst different genotypes needed to be compared, identical magnification, resolution, laser power, and detector settings were used.

## Sequence analysis

Analysis of the probability that each amino acid of Jub is within an intrinsically disordered region was performed using IUPred3 [44]. Sequence comparisons between different LIM domains were performed using pairwise Blastp.

## Image quantitation and analysis

To measure wing sizes, individual adult wings were mounted on slides, photographed, and then traced manually and areas measured using Fiji software [54].

Quantification of *ex-lacZ* was performed using Volocity software (Perkin Elmer), with DNA staining used to define nuclear volumes to be compared. Anterior and posterior wing pouch regions were traced manually, using *hh-Gal4* to define the posterior region and folds to define the wing pouch. The mean intensity per pixel was calculated for each compartment, and the posterior-to-anterior ratio calculated for each wing disc.

Quantification of Wts:V5 was performed using Fiji software, with Ecad staining used to define junctions. Prior to quantitation images were processed using ImSANE [55] to define the apical junctional surface, which compensates for the curvature of the disc and removes peripodial signal. Ecad signal was then used to create an ROI, and separately hh-Gal4 signal was used to define A and P ROIs. Wts:V5 intensities within Ecad and A, and within Ecad and P, were then measured and a P/A ratio calculated for each disc.

To quantify cell shapes, we used the Tissue Analyzer Fiji plugin [56] on flattened images of Ecad signal to segment individual cells within the P compartment of the wing pouch. Cell eccentricity was then calculated using Quantify Polarity software [57]. This program takes segmented cells as input and outputs descriptions of various cell features, including eccentricity.

## Plasmids

Plasmids encoding Jub constructs with a C-terminal msVenus-tag were a gift from J. Zallen [15]. FLAG-tagged versions were created by subcloning from these from UASp-msVenus into pUAST-attB-Flag. Amino acids of Jub included are: Jub-FL (1–718), Jub-preLIM (1–505), Jub LIM only (506–718), Jub preLIM+LIM1 (1–559 and 693–718), Jub preLIM+LIM2 (1–505, 571–623, and 693–718), Jub preLIM+LIM3 (1–505 and 631–718), Jub preLIM+LIM12 (1–623 and 693–718), Jub preLIM+LIM23(1–505, 571–718), Jub preLIM+LIM13 (1–570 and 631–718). UAS-Step-Flag-HA was obtained from the *Drosophila* Genomics Research Center and Step was isolated by PCR, using primers Steppke_fwd, `CTACTAGTCCAGTGTGGTGGATG GC ATCCCTCCATCAG` and Steppke_rvs, `ACTGTGCTGGATATCTGCAGTAA CTCTTGCTG AGTGCC` and subcloned into pAC5.1-V5-His to create pAC5.1-Step-V5-His. Step-V5-His from pAC5.1-Step-V5-His was cloned into pUAST-attB to create pUAST-attB-Step-V5-His, which was used for co-immunoprecipitation experiments. Additional plasmids used were pAw-Gal4, pAC5.1-Wts-V5-His [36] and pUAS-α-CatN2:V5 [20].

## Cell culture, immunoprecipitation and western blotting

S2 cells were cultured in Schneider's *Drosophila* Medium (Gibco), supplemented with 10% FBS and antibiotics, at 28˚C. Cells were transfected using Effectene (Qiagen), incubated for 40–48 hours at 28˚C and then lysed in RIPA buffer (140mM NaCl, 10mM Tris-HCl pH 8.0, 1mM EDTA, 1.0% Triton X-100, 0.1% SDS, and 0.1% sodium deoxycholate) with protease inhibitor and phosphatase inhibitor for 30 minutes at 4˚C. Cell debris was pelleted and 30μL of cell lysate was saved for input samples. Lysates were then pre-cleared using 25 μL of Protein A agarose beads (Pierce) and then incubated with 25 μL of mouse anti-V5 agarose beads (Sigma) and washed with RIPA buffer. Samples were run on SDS PAGE gels (BioRad) and transferred to a nitrocellulose membrane using Trans-Blot®Turbo™ (Bio-Rad). Membrane was incubated with primary antibodies mouse anti-V5 (Invitrogen, 1:5000), rabbit anti-FLAG (Sigma-Aldrich, 1:2,000), mouse anti-GAPDH (1:10,000; Imgenex) overnight at 4˚C, and secondary antibodies anti-mouse IgG-800 (LI-COR Biosciences, 1:10,000) and anti-rabbit IgG-680 (LI-COR Biosciences, 1:10,000) for 2 h at room temperature and then imaged using a Li-Cor Odyssey CLX. Bands were quantified using LiCor ImageStudio software.

## Statistical analysis

Statistical tests were performed using GraphPad software (Prism). For comparison of image and western blot signal intensities we performed one-way ANOVA on the log transform of the ratio with Tukey for multiple comparisons. Comparison of wing areas used one-way ANOVA with Tukey for multiple comparisons. Distributions of cell shapes were compared using a Kruskal-Wallis test with Dunn's test for multiple comparisons. For all statistical tests, ns indicates $P > 0.05$, * indicates $P \leq 0.05$, ** indicates $P \leq 0.01$, *** indicates $P \leq 0.001$, **** indicates $P \leq 0.0001$.

## Supporting information

**S1 Fig. The Jub preLIM regions is predicted to be intrinsically disordered.** The probability that each residue in Jub is part of an intrinsically disordered region, as calculated using IUPred3 [44], is displayed. The positions of the three LIM domains are also indicated. (TIF)

**S2 Fig. Additional characterization of Yki target gene expression.** A-D) Representative wing discs from larvae expressing *ex-lacZ* (magenta), *hh-Gal4 UAS-Dcr2* and A) control B)

*UAS-RNAi-jub UAS-Jub-preLIM+1:msVenus.* C) *UAS-RNAi jub UAS-Jub-preLIM+2:msVenus.* D) *UAS-RNAi jub UAS-Jub-preLIM+3:msVenus.* All discs are shown at the same magnification (scale bar = 40 μm). E) Representative wing disc stained for Diap1 (red/white) from larvae expressing *hh-Gal4 UAS-Dcr2*, shown at the same magnification as for *ex-lacZ* stains. (TIF)

**S1 Raw images. Raw images of whole blots for Figs 1J, 5A and 5B.** (PDF)

## Acknowledgments

We thank J. Zallen, B. Hay, the *Drosophila* Genomics Research Center, and the Bloomington stock center for plasmids, antibodies and *Drosophila* stocks, N. Qureshi for assistance with examination of Wts localization images, and B. Tripathi for assistance with image segmentation and analysis.

## Author Contributions

**Conceptualization:** Cordelia Rauskolb, Kenneth D. Irvine.

**Formal analysis:** Cordelia Rauskolb, Kenneth D. Irvine.

**Funding acquisition:** Kenneth D. Irvine.

**Investigation:** Cordelia Rauskolb, Ahri Han, Elmira Kirichenko.

**Methodology:** Cordelia Rauskolb, Ahri Han, Elmira Kirichenko, Consuelo Ibar.

**Project administration:** Kenneth D. Irvine.

**Supervision:** Cordelia Rauskolb, Consuelo Ibar, Kenneth D. Irvine.

**Writing – original draft:** Kenneth D. Irvine.

**Writing – review & editing:** Cordelia Rauskolb, Ahri Han, Elmira Kirichenko, Consuelo Ibar, Kenneth D. Irvine.

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
