## [Decision Letter · Decision Letter 0]

28 Jun 2022

PONE-D-22-14050Analysis of the Drosophila Ajuba LIM protein defines functions for distinct LIM domainsPLOS ONE

Dear Dr. Irvine,

Your manuscript has now been reviewed by an external referee and by me. We both find that it is a very pretty and thorough analysis of the domain structure of Jub and its consequences in the wing. The referee raises several sensible points that should be addressed, so I will request minor revision, which will not require re-review. Please be sure that your revised manuscript responds to *all* the referee comments. Some key points are listed below.The referee would like you to comment on the apparent differences between the localization observed with single-LIM constructs in the wing vs published data in the embryo. You are welcome to take the suggestion of the referee and test a few key constructs in the embryo in your own hands to see whether that clarifies the situation, but that is just a suggestion, not a condition of publication. Similarly, regarding Fig 1 G-I, it would be helpful to comment (and perhaps test) whether expression level is reduced for the single-LIM constructs or whether the images simply give that appearance due to lack of localization.The referee asks for further information about the differences between jub and LIMD. Anything you can provide would certainly be interesting and worthwhile, but we do not require an experimental analysis of LIMD as a condition of publication.Regarding quantification of eccentricity in Fig 6F, it would be helpful also to do the KW test of the double-LIM constructs relative to RNAi, to ask whether 12 and 23 give any significant rescue, in addition to the test relative to FL, which queries completeness of rescue. Slightly more thorough explanation of how this analysis is interpreted would also be helpful. The referee also offers some interesting ideas about the potential link between eccentricity and myosin activity. These again are at the discretion of the authors; such experiments are not required though the thoughts of the authors would undoubtedly be illuminating.

We look forward to receiving your revised manuscript.

Kind regards,

Edward Giniger

Academic Editor

PLOS ONE

Journal Requirements:

[This research was supported by National Institutes of Health grant GM131748 (KDI).]

 [This research was supported by National Institutes of Health (www.nih.gov) grant GM131748 to KDI. 

The funders had no role in study design, data collection and analysis, decision to publish, or preparation of the manuscript.]

Reviewers' comments:

Reviewer's Responses to Questions

**Comments to the Author**

1. Is the manuscript technically sound, and do the data support the conclusions?

Reviewer #1: Yes

2. Has the statistical analysis been performed appropriately and rigorously? 

Reviewer #1: Yes

3. Have the authors made all data underlying the findings in their manuscript fully available?

Reviewer #1: Yes

4. Is the manuscript presented in an intelligible fashion and written in standard English?

Reviewer #1: Yes

5. Review Comments to the Author

Reviewer #1: This MS analyses the requirement for the various LIM domains in the function of Jub in regulating in the wing disc: a-catenin interaction, cell shape, Step localisation to AJ, wing growth, and Wts activity (through Yki targets) and localisation to AJ. In a clear and efficient way, they find that the functional contributions of the three LIM domains have distinct roles, and the abstract nicely summarises those clear findings. The fact that the jub LIM domains have distinct roles was already shown in embryos for some of the jub functions, specially related to catenin, myosin activity and cell shape, but not for its function in wts regulation.

The authors find that the way multiple LIM domains are required for significant junctional localisation in wing discs differs from studies in Drosophila embryos, where ‘constructs with a single LIM domain, and even a preLIM-only construct, exhibited some junctional localization [15’]. Could the authors explore these differences? Checking their conditions in embryos? Specially since their single LIM constructs did not seem to express very much (see fig1G-I…could the differences be because the wing disc localisation was studied in the background of the jub RNAi? And related to this, could the authors investigate if the single LIM constructs in fig 1G-I are expressing reasonably well? By WB for example?

Another point that needs to be resolved is the differences between jub and LIMD1…are they real differences between the 2 proteins, or alternatively might stem from differences in experimental conditions? This could be tested by the authors studying both jub and LIMD1 constructs in their system.

Is the extremely efficient co-ipp of prelim+23 with catenin anything that seems to happen in vivo? Do the authors see higher levels of this construct in AJs? Did not seem so to me. Is this reflecting a function of LIM1 in regulating a functional interaction of jub with catenin?

There is a nice correlation on the wing growth, yki targets and wts localisation to AJ. Ijust have a question: are there higher levels of junctional wts in the jub-FL?

Fig5A wts bands: they don’t all seem to have the exact same size…maybe PTM when coexpressed with some Jub constructs? Or just a technical issue?

prelim+23 rescues cell shape? I find Fig6F difficult to understand. The colour codes are not completely matching. Is the rescue a conclusion from having more cells with an eccentricity value closer to jubFL? But the value is higher, how is this quantified? M&M says: Cell

eccentricity was then calculated using Quantify Polarity software…could the authors explain more? Also, regarding cell shape, it would be nice if the authors could explore more the link between myosin and their constructs, as myo is known to be linked to ajuba to and cell shape in embryos. Further think of ways to quantify myo acitivty? Or genetic interactions between the actomyo pathway and the jub constructs?

6. PLOS authors have the option to publish the peer review history of their article (what does this mean?). If published, this will include your full peer review and any attached files.

Reviewer #1: No

---

## [Author Response · Author response to Decision Letter 0]

20 Jul 2022

Response to Editor’s and Referee’s comments

We thank both the Editor and Referee for their thoughtful comments and positive response to our manuscript.

Response to Editor’s comments

The referee would like you to comment on the apparent differences between the localization observed with single-LIM constructs in the wing vs published data in the embryo. You are welcome to take the suggestion of the referee and test a few key constructs in the embryo in your own hands to see whether that clarifies the situation, but that is just a suggestion, not a condition of publication. Similarly, regarding Fig 1 G-I, it would be helpful to comment (and perhaps test) whether expression level is reduced for the single-LIM constructs or whether the images simply give that appearance due to lack of localization.

Response: As noted below in the response to Reviewer’s, we don’t think it makes sense for us to repeat others’ published experiments. We have now added a comment noting that we think the reduced signal for the single LIM domain constructs most likely reflects mis-localization. 

The referee asks for further information about the differences between jub and LIMD. Anything you can provide would certainly be interesting and worthwhile, but we do not require an experimental analysis of LIMD as a condition of publication.

Response: Thanks, as noted in response to reviewer below, we do not want to do additional experiments here.

Regarding quantification of eccentricity in Fig 6F, it would be helpful also to do the KW test of the double-LIM constructs relative to RNAi, to ask whether 12 and 23 give any significant rescue, in addition to the test relative to FL, which queries completeness of rescue. Slightly more thorough explanation of how this analysis is interpreted would also be helpful. The referee also offers some interesting ideas about the potential link between eccentricity and myosin activity. These again are at the discretion of the authors; such experiments are not required though the thoughts of the authors would undoubtedly be illuminating.

Response: We have now added this additional KW test to Fig 6F, as requested, and we also added a more detailed explanation of the analysis.

Response to Reviewer’s comments

Reviewer #1: This MS analyses the requireme nt for the various LIM domains in the function of Jub in regulating in the wing disc: a-catenin interaction, cell shape, Step localisation to AJ, wing growth, and Wts activity (through Yki targets) and localisation to AJ. In a clear and efficient way, they find that the functional contributions of the three LIM domains have distinct roles, and the abstract nicely summarises those clear findings. The fact that the jub LIM domains have distinct roles was already shown in embryos for some of the jub functions, specially related to catenin, myosin activity and cell shape, but not for its function in wts regulation.

The authors find that the way multiple LIM domains are required for significant junctional localisation in wing discs differs from studies in Drosophila embryos, where ‘constructs with a single LIM domain, and even a preLIM-only construct, exhibited some junctional localization [15’]. Could the authors explore these differences? Checking their conditions in embryos? Specially since their single LIM constructs did not seem to express very much (see fig1G-I…could the differences be because the wing disc localisation was studied in the background of the jub RNAi? And related to this, could the authors investigate if the single LIM constructs in fig 1G-I are expressing reasonably well? By WB for example?

Response: The constructs we used to analyze localization in wing discs were obtained from the Zallen lab, who did the cited studies in the embryo. Consequently, we wouldn’t expect to obtain results in the embryo any different from what the Zallen lab reported. It’s possible that there are differences between embryonic and wing disc junctions. Our focus was to identify functional differences between Jub constructs, which we assayed by rescuing activity, so we think it’s important that the localization be reported in jub RNAi backgrounds. It's possible there are some differences in stability between the different isoforms, but the decreased signal is most likely due primarily to the mis-localization – instead of being concentrated at junctions, the single LIM Jub constructs are spread throughout the cytoplasm. However, even if the levels are lower for the single LIM domain constructs it wouldn’t change any of our conclusions, so we prefer not do additional experiments on this.

Another point that needs to be resolved is the differences between jub and LIMD1…are they real differences between the 2 proteins, or alternatively might stem from differences in experimental conditions? This could be tested by the authors studying both jub and LIMD1 constructs in their system.

Response: The Longmore lab did some studies of LIMD1 in Drosophila wing discs, which is the same system. Where there are differences they could be real differences between proteins, or differences in experimental analysis, but we don’t think it makes sense for us to repeat the already published experiments of another group.

Is the extremely efficient co-ipp of prelim+23 with catenin anything that seems to happen in vivo? Do the authors see higher levels of this construct in AJs? Did not seem so to me. Is this reflecting a function of LIM1 in regulating a functional interaction of jub with catenin?

Response: The higher co-IP in S2 cells with this construct is not reflected in higher junctional localization in vivo. We are not sure what explains the difference, but there are some key differences between the in vivo and in vitro experiments, including protein expression levels, existence of AJ in wing discs but not in S2 cells (so interactions happen in a different context), and use of full length a-catenin in vivo versus the Jub binding fragment in S2 cells. We modified the text to note these possibilities.

There is a nice correlation on the wing growth, yki targets and wts localisation to AJ. Ijust have a question: are there higher levels of junctional wts in the jub-FL?

Response: Quantitation of the Posterior/Anterior ratio for junctional Wts signal shows that it is 10% higher in Posterior cells (where Jub-FL is over-expressed in these experiments) than in Anterior cells, so most likely there is a slight effect (by eye Wts levels appear similar in anterior and posterior cells without Jub-FL over-expression). However, as the focus for this experiment is comparison between different Jub constructs, we don’t think it makes sense to include this minor point, which would require additional experiments for rigorous quantitation.

Fig5A wts bands: they don’t all seem to have the exact same size…maybe PTM when coexpressed with some Jub constructs? Or just a technical issue?

Response: Indeed there may be some PTM, and we are currently investigating this possibility, but that is well beyond the scope of this manuscript.

prelim+23 rescues cell shape? I find Fig6F difficult to understand. The colour codes are not completely matching. Is the rescue a conclusion from having more cells with an eccentricity value closer to jubFL? But the value is higher, how is this quantified? M&M says: Cell

eccentricity was then calculated using Quantify Polarity software…could the authors explain more? Also, regarding cell shape, it would be nice if the authors could explore more the link between myosin and their constructs, as myo is known to be linked to ajuba to and cell shape in embryos. Further think of ways to quantify myo acitivty? Or genetic interactions between the actomyo pathway and the jub constructs?

Response: The analysis we did compares the distribution of cell eccentricity values. Eccentricity is a geometric description of shape, formally Eccentricity (e) = c/a Where, c = distance from the center to the focus; a = distance from the center to the vertex. A circle has an eccentricity of 0, a line has an eccentricity of 1, and an ellipse has an eccentricity between 0 and 1, with more elongated ellipses having higher values (ie, closer to 1). We used a freely available software package that takes segmented cells as input and outputs descriptions of various cell features, including eccentricity. We then compared the distribution of cell eccentricities between the positive control (Jub-FL) and the other genotypes, which revealed that the distribution of cell eccentricity for LIM23 was not significantly different from Jub-FL, whereas the distribution of cell eccentricities for the other genotypes was significantly different from Jub-FL. This comparison was done using a Kruskal-Wallis test with Dunn’s test for multiple comparisons. This is a statistical test that compares distributions of values among multiple samples. It’s a non-parametric method (does not require a normal distribution). The comparison evaluates the entire distribution, not simply peak or mean values. We have revised the text to provide a more detailed explanation of this analysis. We have also now, as suggested by the editor, added the K-W test for comparisons of the different Jub constructs to the jub RNAi condition to Fig. 6F.

We have described links between myosin and Jub in previous publications, including the discovery that effects of Jub on myosin are mediated by effects on Step. Thus, we have included some exploration of how the Jub constructs influence myosin through our analysis of interactions with and effects on Step, as presented in Figs 5 and 7. At this point, we don’t think exploring genetic interactions between Jub and myosin or the “actomyo pathway” would add anything.

---

## [Editor Report · Decision Letter 1]

2 Aug 2022

Analysis of the Drosophila Ajuba LIM protein defines functions for distinct LIM domains

PONE-D-22-14050R1

Dear Dr. Irvine,

We’re pleased to inform you that your manuscript has been judged scientifically suitable for publication and will be formally accepted for publication once it meets all outstanding technical requirements.

Kind regards,

Edward Giniger

Academic Editor

PLOS ONE
---

## [Editor Report · Acceptance letter]

4 Aug 2022

PONE-D-22-14050R1 

Analysis of the *Drosophila* Ajuba LIM protein defines functions for distinct LIM domains 

Dear Dr. Irvine:

I'm pleased to inform you that your manuscript has been deemed suitable for publication in PLOS ONE. Congratulations! Your manuscript is now with our production department. 

Kind regards, 

on behalf of

Dr. Edward Giniger 

Academic Editor

PLOS ONE